# Implementation of Parallel Cascade Identification at Various Phases for Integrated Navigation System

Umar Iqbal [1,*], Ashraf Abosekeen [2,*], Jacques Georgy [3], Areejah Umar [4], Aboelmagd Noureldin [5] and Michael J. Korenberg [6]

1 Department of Electrical and Computer Engineering, Mississippi State University, Starkville, MS 39759, USA
2 Electrical Engineering Branch, Military Technical College (MTC), Cairo 11762, Egypt
3 TDK-InvenSense Inc., #405, 1000 Veterans Place NW, Calgary, AB T3B 4M1, Canada; jacquesfw@gmail.com
4 Department of Biological Science, Mississippi State University, Starkville, MS 39759, USA; au147@msstate.edu
5 Department of Electrical and Computer Engineering, Royal Military College of Canada (RMCC), Kingston, ON K7K7B4, Canada; Aboelmagd.Noureldin@rmc.ca
6 Department of Electrical and Computer Engineering, Queen's University, Kingston, ON K7L3N6, Canada; korenber@queensu.ca
* Correspondence: umar@ece.msstate.edu (U.I.); a.abosekeen@ieee.org (A.A.)

**Abstract:** Global navigation satellite systems (GNSS) are widely used for the navigation of land vehicles. However, the positioning accuracy of GNSS, such as the global positioning system (GPS), deteriorates in urban areas due to signal blockage and multipath effects. GNSS can be integrated with a micro-electro-mechanical system (MEMS)–based inertial navigation system (INS), such as a reduced inertial sensor system (RISS) using a Kalman filter (KF) to enhance the performance of the integrated navigation solution in GNSS challenging environments. The linearized KF cannot model the low-cost and small-size sensors due to relatively high noise levels and compound error characteristics. This paper reviews two approaches to employing parallel cascade identification (PCI), a non-linear system identification technique, augmented with KF to enhance the navigational solution. First, PCI models azimuth errors for a loosely coupled 2D RISS integrated system with GNSS to obtain a navigation solution. The experimental results demonstrated that PCI improved the integrated 2D RISS/GNSS performance by modeling linear, non-linear, and other residual azimuth errors. For the second scenario, PCI is utilized for modeling residual pseudorange correlated errors of a KF-based tightly coupled RISS/GNSS navigation solution. Experimental results have shown that PCI enhances the performance of the tightly coupled KF by modeling the non-linear pseudorange errors to provide an enhanced and more reliable solution. For the first algorithm, the results demonstrated that PCI can enhance the performance by 77% as compared to the KF solution during the GNSS outages. For the second algorithm, the performance improvement for the proposed PCI technique during the availability of three satellites was 39% compared to the KF solution.

**Keywords:** land vehicle navigation; system identification; inertial sensors; GNSS; Kalman filter; parallel cascade identification

## 1. Introduction

System identification began by the middle of the twentieth century, and it is highly dependent on its purpose and application [1,2]. It can be used for control strategies or to analyze the properties of a system. System identification is utilized in a variety of applications to address the modeling problems of dynamic systems. The application of the system identification technique plays a vital role in deciding whether a crude model will be enough or if an accurate model is required for the system dynamics. It is also possible to model the environment of the system to address the application need [3,4]. Linear system identification has played a vital role in the development of modern design methods [3–5]. Linear system identification methods include least-squares identification of a parametric

model, repeated least squares, correlated residuals, the maximum likelihood method, the Tally principle, and Levin's method. System identification requires the following steps:

1. Input/output data measurement with appropriate sampling procedures either in the time domain or in the frequency domain.
2. A set of candidate models and to choose a suitable model structure.
3. An estimation method for minimization of fit between model (predicted) output and measured output.

The mathematical representation of a system's dynamics is termed modeling. Modeling of the dynamical system is more challenging as the effects of actions take some time to occur. A single system can be described by different models depending upon its applications. A black box approach is based entirely on observed inputs and outputs of the system, as shown in Figure 1. It is widely used for many engineering problems. Using this approach, we can decompose a system into different modules. It is very suitable for linear, time-invariant systems and can also be applied to non-linear systems. However, linear system identification is not able to address many practical time-varying systems, and it becomes necessary to use non-linear system identification techniques [6–9]. The application of non-linear system identification techniques is justified when linear models are not able to handle the excessive non-linear distortion levels. Non-linear system identification techniques include representation of non-linear systems and estimation of a parametric model. For non-linear systems identification, the model selection and parameter estimation are enormously complicated.

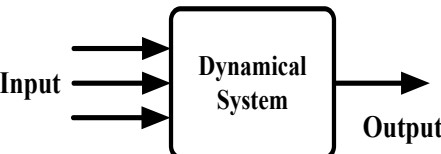

**Figure 1.** Illustration of the input/output block diagram of a system.

This paper reviews the utilization of a non-linear system identification technique called parallel cascade identification (PCI) to improve the overall navigation solution by modeling errors at the sensor and measurement level.

## 2. Overview of Navigation Systems

The last two decades have seen an increasing trend in the use of global navigation satellite systems (GNSS) in a variety of positioning and navigation applications. The GNSS receiver determines the satellite's antenna position, and lever arm compensation is utilized to deduce the receiver position for GPS/RISS integration. GNSS applications include but are not limited to passenger cars, taxis, buses, ambulances, police cars, farming vehicles, fire trucks, and mobile robots [10]. Current navigational systems match the position on the digital map with the help of information from GNSS. Improved digital maps assist in the enhancement of navigational systems [10,11]. Intelligent transport systems (ITS) focus on bringing features like collision warning and mitigation, lane-keeping, lane-changing with route guidance to the desired destination, traffic flow guidance, vulnerable road user detection, driver condition monitoring, and improved vision. These features need navigation systems with higher accuracy and better reliability, availability, and continuity of service [12]. Moreover, GNSS has played a major role in the navigation of unmanned aerial vehicles (UAV), utilizing single or multiple satellite constellations [13]. The researchers utilized a space-based augmentation system (SBAS) to produce an improved positioning for the UAV by 19–22% compared to the single constellation. Furthermore, GNSS, INS, and light direction and ranging (LiDAR) systems have been combined to improve the orientation of low altitude UAVs utilized in coastal applications [14]. The proposed system reduced the orientation error from 0.5 degrees to 0.01 degrees. Although the solution

provided by GNSS is sufficiently accurate (especially when used in differential mode), it is not able to accomplish the requirements of continuity, reliability, and availability. GNSS may suffer from outages, interference, jamming, spoofing, and multipath effects in urban canyons and rural foliage canopies, as shown in Figure 2 [15]. Thus, GNSS alone cannot fulfill the requirements of service for modern navigation systems. In [16], a sidelobe cancellation technique introduced to the multipath effect could be used in GNSS receivers.

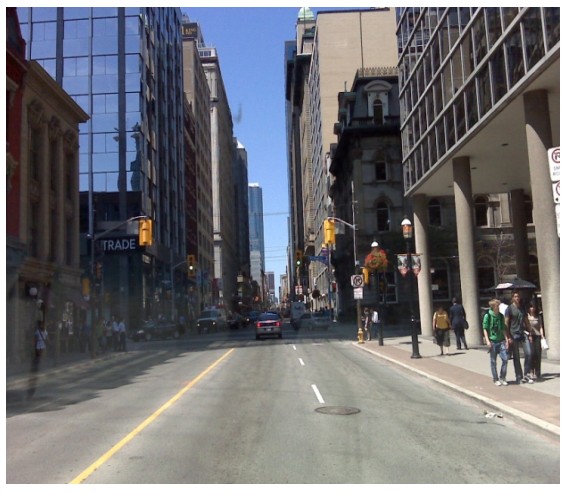

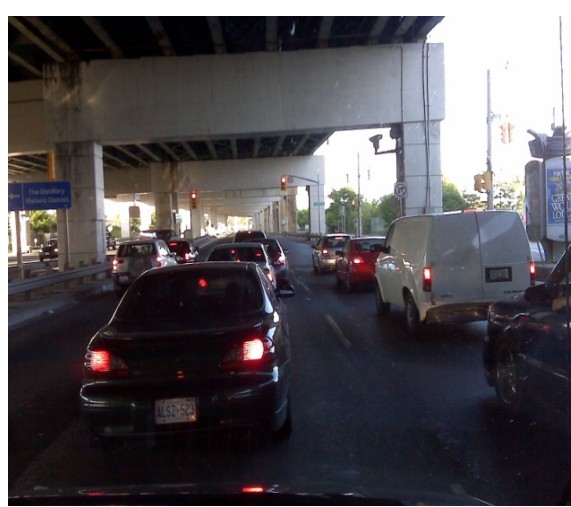

(**a**) Urban Canyon  (**b**) Underpass

**Figure 2.** GNSS in challenging environments.

## 3. Problem Statement

Presently, there is a growing demand for low-cost navigation systems that can provide accurate positioning at all times, even in harsh GNSS environments. The GNSS is considered to be the main navigation system for land vehicle applications; it suffers from several errors that might lead to signal loss or satellite blockages [17–19]. GNSS navigation requires an uninterrupted line of sight (LOS) between the receiver that is mounted on the moving vehicle and a minimum number of four satellites. Some environments such as urban canyons do not allow this primary condition to be accomplished. Consequently, the receiver will not be able to produce a solution. Therefore, there is a great need for extra sensors/systems mounted on the moving vehicle. The inertial navigation system (INS) is a practical option as a backup navigation system [20–22]. The central contributing unit in the INS is the inertial measuring unit (IMU). A full IMU is a self-contained device that consists of three accelerometers and three gyroscopes to continuously measure three orthogonal linear accelerations and three orthogonal angular rates, respectively. These raw measurements are then integrated in a strap-down manner to provide position, velocity, and altitude using a sequence of mechanization equations [23,24].

Thus, INS provides a continuous navigation solution, but navigational errors grow exponentially with time due to several factors that may include sensor bias, drift, misalignment, and scale factor instability. Other complementary navigation methods such as steering encoder, odometer, velocity encoder, and electronic compass can be used to curtail the error growth rate of the INS. A more accurate navigational solution can be obtained by integrating these motion sensors with GNSS [25,26].

Traditionally, the integration of GNSS with other systems like INS has been provided by KF or extended KF (EKF), which relies on a linearized error model of both GNSS and INS. KF has provided a reliable GNSS/INS integration solution for high-end navigational and tactical grade INS. However, KF may not be able to address the complex stochastic and high order errors of MEMS grade sensors. This will result in large values of the non-linear error terms, which are usually ignored during the linearization process while generating the error model for KF. The error models are required to analyze and estimate different

error sources associated with the proposed RISS. The error state vector for RISS includes coordinate errors, velocity errors, azimuth error, stochastic error in odometer derived acceleration, stochastic error in the accelerometers, and gyroscope reading. When the low-cost MEMS IMU is integrated with GNSS by using traditional KF integration techniques, the solution becomes unreliable, especially in prolonged GNSS outages. It is vital to have accurate error models to achieve consistent KF results. However, it is challenging to model MEMS sensors as they have composite error characteristics [23–27]. The lack of rigorous error models can diminish the overall navigation accuracy.

The main objective of this paper is to enhance the performance of integrated MEMS-based INS/GNSS navigation systems through the PCI non-linear modeling approach that can deal with the non-linear parts of INS and GNSS errors. In order to achieve this objective, this paper aims at the following:

1. A review of the PCI algorithm, a non-linear system identification technique with the details of different implementation steps, is discussed.
2. The research approach in this paper relies on reduced inertial sensor systems (RISS), which limits the reliance on MEMS-based gyroscopes to avoid their high levels of noise and drift rates. The RISS incorporating single-axis gyroscope, vehicle odometer, and accelerometers will be considered for the integration with GNSS in one of two schemes:

    (a) Loosely coupled where GNSS position and velocity are used for the integration.
    (b) Tightly coupled where GNSS pseudorange and pseudorange rates are utilized.

3. In the first scenario, PCI is employed to enhance the performance of KF by modeling azimuth errors for the RISS/GNSS loosely coupled integration scheme. The azimuth non-linear error model is identified online using PCI, and the corrected azimuth is sent to the KF-based RISS/GNSS integrated module to improve the overall navigation accuracy.
4. Then, PCI is utilized for the modeling of the residual GNSS pseudorange correlated errors. This paper provides a brief review to augment a PCI-based model of GNSS pseudorange correlated errors with a tightly coupled KF, to integrate low-cost MEMS-based RISS and GNSS observations.

## 4. Parallel Cascade Identification

The PCI technique is based on the idea of modeling the non-linear system input/output relation of alternating dynamic linear (L) and static non-linear (N) elements by summing parallel cascades. The model built has a finite number of parallel LN cascade paths, where each path consists of a dynamic linear element followed by a static non-linearity. The static non-linearity can be a polynomial. The model output is the sum of the outputs of the parallel branches, as shown in Figure 3.

Frechet in 1910 proved that in continuous time, a finite memory non-linear system whose output is a continuous mapping of its input can be uniformly approximated over a uniformly bounded equicontinuous set of inputs to an arbitrary degree of precision by a Volterra series of sufficient but finite order [28]. The Volterra series represents a functional expansion of a dynamic, non-linear, time-invariant functional. The Volterra series is commonly used in system identification. Palm [29] showed that any discrete-time Volterra series with limited memory could be uniformly estimated by a limited sum of parallel LNL cascades, where the static non-linearities N are exponentials and logarithmic functions. Korenberg [28] showed that any discrete-time finite memory non-linear system having a finite-order Volterra series representation can be exactly represented by a finite number of parallel LN cascade paths, where the N are polynomials. In practice, additional LN elements may be added in any cascade path to increase accuracy [28]. A major benefit of this technique is its independence of a Gaussian or white input, but it identifies separate L and N elements and may change depending on the statistical properties of the input chosen [28]. One cascade can be found at a time, and the non-linearities in the models are localized in static functions. This reduces the computation, as higher-order non-linearities

are approximated using higher degree polynomials in the cascades rather than higher-order kernels in a Volterra series approximation. The technique begins by estimating the non-linear system by a first such cascade. The residual (i.e., the difference between the system output and the cascade outputs) is treated as the output of a new non-linear system; a second cascade is found to estimate the latter system, and thus the parallel array can augment one cascade at a time. Consider an unknown dynamic non-linear system with accessible input $x(n)$ and output $y(n)$ where $n = 0, \ldots \ldots, T$, $T$ is the length of the dataset or record used for the training. Under wide conditions, one can model the system using parallel cascade supposing that the output can depend on delayed input values $x(n - j)$, for $j = 0, \ldots \ldots, R$. Here $R$ is the maximum lag or delay and $(R + 1)$ is the memory length (since the series output $y(n)$ depends on input delays from 0 to $R$ lags or delay).

The maximum degree of non-linearity required for a good approximation of the system is $D$. The polynomial degree $D$ cannot exceed $(T - R)$ since there are $(D + 1)$ coefficients to estimate in the polynomial, and there would be exactly $(T - R + 1)$ points available for the estimation. However, a much smaller value is in practice used for the polynomial degree $D$, and its value is application dependent. Figure 4 shows the main steps of the PCI algorithm.

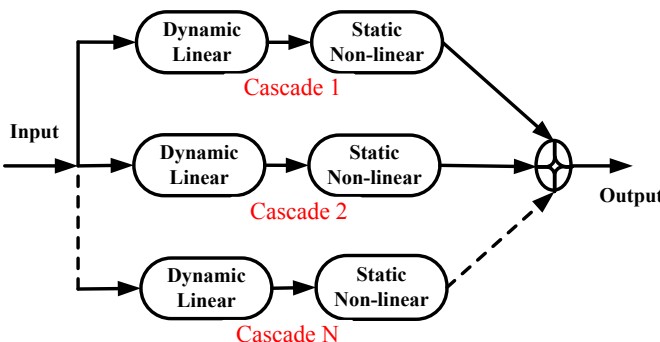

**Figure 3.** Illustration of the parallel cascade identification.

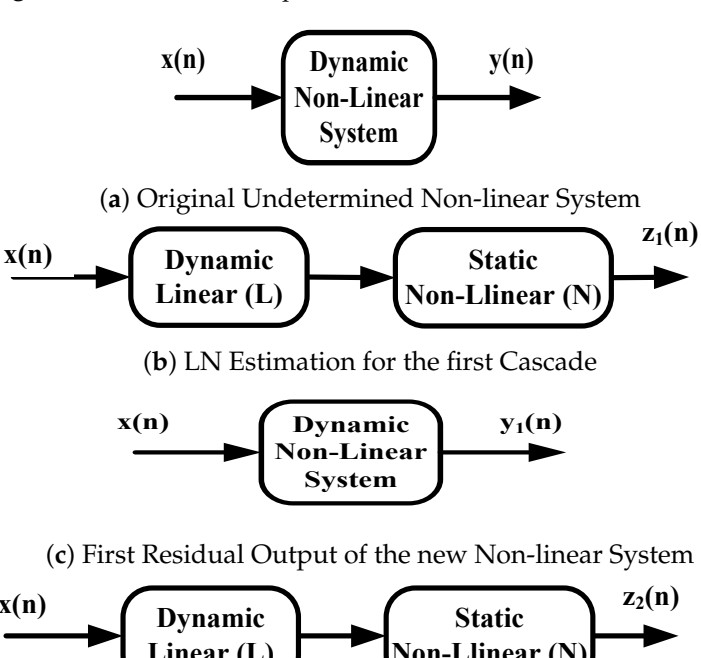

(**a**) Original Undetermined Non-linear System

(**b**) LN Estimation for the first Cascade

(**c**) First Residual Output of the new Non-linear System

(**d**) Second LN Cascade

**Figure 4.** Step-by-step implementation of PCI algorithm.

The PCI technique can be explained in the following five steps:

1. The first cascade output of the non-linear dynamic system is $z1(n)$, as shown in Figure 4b, and it is estimated by a cascade of a dynamic linear $(L)$ followed by a static non-linear $(N)$ element.
2. Then, compute the first residual as shown in Figure 4c.

$$y_1(n) = y(n) - z_1(n) \tag{1}$$

3. Figure 4d shows the estimation of the new non-linear system having input $x(n)$ and output $y1(n)$ by a cascade of $L2$ followed by $N2$.
4. Compute the second residual.

$$y_2(n) = y_1(n) - z_2(n) \tag{2}$$

5. And so on …
   Let $y_k(n)$ be the residual after fitting the $k$-th cascade, so $y_0(n) = y(n)$. Let $z_k(n)$ be the output of the $k$-th cascade, so

$$y_k(n) = y_{k-1}(n) - z_k(n); \qquad where \quad k = 1, 2, 3, \ldots \tag{3}$$

*Details of the PCI Algorithm*

The salient steps for obtaining the impulse response of the dynamic linear element for the current cascade can be listed as follows. When identifying the $k$-th cascade, the existing residual before the addition of the $k$-th cascade is $y_{k-1}(n)$. The approach utilized in this paper to obtain the impulse response $g_k(j)$ of the linear element $L_k$ of the $k$-th cascade uses cross-correlations of the input with the current residual, and this impulse response will be one of the following:

- Impulse response will be input residual cross-correlation:

$$g_k(j) = \phi_{xy_{k-1}}(j) = \frac{1}{T - R + 1} \sum_{n=R}^{T} y_{k-1}(n)x(n-j) \qquad j = 0, \ldots, R \tag{4}$$

  A portion of second order cross-correlations of input and residual $\phi_{xxy_{k-1}}(j,A)$ is used; thus, the impulse response will be as follows:

$$g_k(j) = \phi_{xxy_{k-1}}(j, A) \pm c\delta(j - A) \tag{5}$$

  where $\delta(.)$ is the Kronecker delta function, the sign is chosen at random, $A$ is chosen at random from $0, \ldots, R$, and $c$ is chosen such that $c \to 0$ as $\overline{y_{k-1}^2}(n) \to 0$, e.g., $c = \frac{\overline{y_{k-1}^2}(n)}{\overline{y^2}(n)}$ (here, the over-bar means the finite-time average from $n = R$ to $n = T$ as in the expression for $\phi_{xy_{k-1}}(j)$ immediately above).

- A portion of the third order input residual cross-correlation $\phi_{xxxy_{k-1}}(j,A_1,A_2)$ will be used; thus, the impulse response will be as follows:

$$g_k(j) = \phi_{xxxy_{k-1}}(j, A_1, A_2) \pm c_1\delta(j - A_1) \pm c_2\delta(j - A_2) \tag{6}$$

- We can use this expression up until the "$n$" order cross-correlation using the following:

$$g_k(j) = \phi_{x\ldots xy_{k-1}}(j, A_1, \ldots, A_{D-1}) \pm c_1\delta(j - A_1) \pm \cdots \pm c_{D-1}\delta(j - A_{D-1}) \tag{7}$$

Nevertheless, in practice, cross-correlations up to the third order are typically enough. The output of the linear element calculated by convolution summation is as follows:

$$u_k(n) = \sum_{j=0}^{R} g_k(j)x(n-j) \tag{8}$$

Here, the linear element's output $u_k(n)$ depends on input values $x(n), x(n-1), \ldots, x(j-R)$, linear elements have the memory length of $R+1$, and $g_k(j)$ is the impulse response of the linear element $L_k$ at beginning the $k$-th cascade.

To obtain the static non-linear element for the current cascade by polynomial fitting, the following steps are followed. First $\overline{u_i^2(n)}$ is calculated. Let it equal $M$, and then the impulse response of the dynamic linear element is adjusted to be $\tilde{g}_i(n) = \frac{g_i(j)}{\sqrt{M}}$ to ensure that $\overline{u_i^2(n)} = 1$.

A polynomial (static non-linearity) is best fit to minimize the mean square error (MSE) of the approximation of the residual. To fit the static non-linearity, the coefficient aids $(d = 0, \ldots, D)$ are found to minimize.

$$e_i = \overline{\left(y_{i-1}(n) - \sum_{d=0}^{D} a_{id} u_i^d(n)\right)^2} = \frac{1}{T-R+1} \sum_{n=R}^{T} \left(y_{i-1}(n) - \sum_{d=0}^{D} a_{id} u_i^d(n)\right)^2 \quad (9)$$

As noted, the over-bar here means a finite-time average. Minimizing $e_i$ with respect to each of the polynomial coefficients leads to $D+1$ equations in $D+1$ unknowns "$a_{id}$".

$$\overline{y_{i-1}(n)u_i^q(n)} = \sum_{d=0}^{D} a_{id} \overline{u_i^{d+q}(n)} \qquad where, \quad q = 0, 1, \ldots, D \quad (10)$$

It is important to know whether it is suitable to add the current cascade to the built model or not. The new cascades are to minimize the mean-square error such as to drive the cross-correlations of the input with the residual to zero [28,30] and are given by the following equation:

$$\overline{z_k^2(n)} > \frac{4}{T-R+1} \overline{y_{k-1}^2(n)} \quad (11)$$

where $\overline{z_k^2(n)}$ denotes the mean square of the candidate cascade's output, and $\overline{y_{k-1}^2(n)}$ denotes the mean square of the current residual, i.e., the residual remaining from the cascades already present in the model.

The following are four stopping conditions of building a parallel cascade for the PCI algorithm [30].

1.　When a certain number of cascades are added;
2.　When a certain number of cascades are analyzed (whether they are included or rejected);
3.　When MSE is adequately insignificant;
4.　When no residual candidate cascade can reduce the MSE considerably.

## 5. The 2D Reduced Inertial Sensor System

Two-dimensional RISS was proposed in [23,24] involving a single-axis gyroscope and a speed sensor to provide a full 2D positioning solution. The overview of the RISS mechanization can be seen in Figure 5.

For RISS mechanization, the azimuth angle is acquired by integrating the gyroscope measurement $\omega_z$. As this measurement includes the Earth's rotation component and rotation of the local-level frame on the Earth's curvature, these quantities are removed from the measurement before integration [18,19,23,24]. Assuming a relatively small pitch angle for land vehicle applications, the rate of change of the azimuth angle directly in the local-level frame is as follows:

$$\dot{A} = -\left(\omega_z - b_z - \omega^e \sin(\varphi) - \frac{v_e \tan(\varphi)}{R_N + h}\right) \quad (12)$$

where $\omega^e$ is the Earth's rotation rate, $b_z$ is the gyro bias, $\varphi$ is the latitude, $v_e$ is the east velocity, $h$ is the altitude of the vehicle, and $R_N$ is the meridian radius of curvature of the Earth.

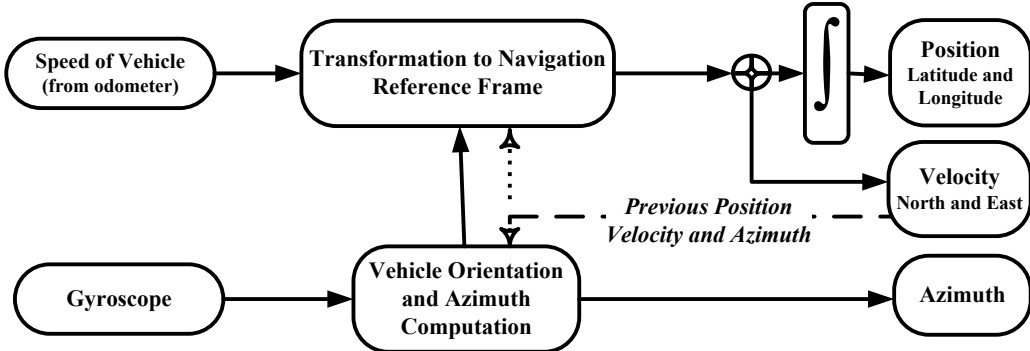

**Figure 5.** Block diagram of 2D RISS.

## 6. The 3D Reduced Inertial Sensor System

The 2D RISS depends on the fact that land vehicles mostly stay on the horizontal plane. Due to the limitation of 2D RISS on roads with slopes, especially in hilly and uneven terrain, 3D RISS [25–27] was developed by incorporating two accelerometers for the provisioning of pitch and roll angles and incorporating the vertical information in the system model to be used by the RISS/odometer/GNSS integration filter. When pitch and roll are calculated from accelerometers, the first integration of gyroscopes to obtain pitch and roll is eliminated; thus, the error in pitch and roll is not proportional to time integration. The outcome of these accurate estimates is superior velocity and position estimates for 3D RISS with odometer, along with upward velocity and altitude that are not calculated before. The overview of the 3D RISS mechanization can be seen in Figure 6.

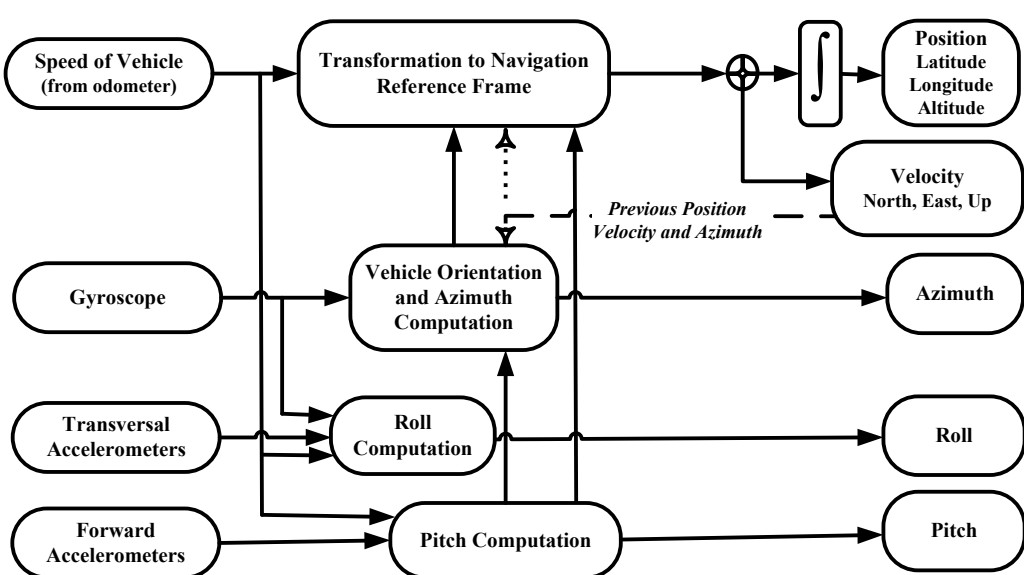

**Figure 6.** Block diagram of 3D RISS.

The pitch angle is derived from the forward acceleration, the acceleration from the odometer, and the gravity as shown in Equation (13).

$$p = sin^{-1}\left(\frac{f_y - a_{od}}{g}\right) \qquad (13)$$

where $p$ is the pitch angle, $f_y$ is the forward accelerometer specific force, $a_{od}$ is the forward acceleration obtained from the odometer, and $g$ is the gravity.

The roll angle is calculated as in Equation (14).

$$r = -sin^{-1}\left(\frac{f_x - v_{od}(\omega_z - b_z)}{gcos(p)}\right) \tag{14}$$

where $f_x$ is the transversal accelerometer specific force, $v_{od}$ is the forward speed , $\omega_z$ is the angular rate, $b_z$ is the gyroscope bias. The azimuth rate $\dot{A}$ is determined from the gyroscope angular rate, the Earth's rotation rate, and the rotation from moving in curvilinear motion by taking in account the latitude and altitude on Earth. The azimuth rate is given by Equation (15).

$$\dot{A} = -\left(\omega_z - b_z - \omega^e sin(\varphi) - \frac{v_e tan(\varphi)}{R_N + h}\right) \tag{15}$$

where $\omega^e$ is the Earth's rotation rate, $b_z$ is the gyro bias, $v_e$ is the east velocity, and $R_N$ is the meridian radius of curvature of the Earth. The 3D velocity components are calculated by projecting the forward speed measured by the odometer using the altitude angles as in Equation (16).

$$v = \begin{bmatrix} v_e \\ v_n \\ v_u \end{bmatrix} = \begin{bmatrix} v_{od}sin(A)cos(p) \\ v_{od}cos(A)cos(p) \\ v_{od}sin(p) \end{bmatrix} \tag{16}$$

where $v_e$, $v_n$, and $v_u$ are the East, North, and Up velocities, respectively.

The 3D position components are obtained from the velocities as in Equation (17), taking the Earth's geometry into consideration:

$$\begin{bmatrix} \dot{\varphi} \\ \dot{\lambda} \\ \dot{h} \end{bmatrix} = \begin{bmatrix} \frac{v_n}{R_N + h} \\ \frac{v_e}{(R_M + h)cos(\varphi)} \\ v_u \end{bmatrix} \tag{17}$$

where $\dot{\varphi}$, $\dot{\lambda}$, and $\dot{h}$ are the latitude, longitude, and altitude rates respectively, and $R_M$ is the normal radius of curvature of the Earth's ellipsoid.

## 7. Kalman Filter

Kalman filtering (KF) is an optimal estimation tool that provides a sequential recursive algorithm for the optimal least mean-variance (LMV) estimation of the system states [31,32]. The theory of KF is well established, and details can be found in [32–34]. KF is the optimal estimator if the system and measurement models are linear. However, the INS/GNSS integration problem has non-linear models. Thus, the linearization of these models is needed, and the filter works with linearized error-state models rather than the total-state non-linear model. Moreover, KF is an algorithm for predicting the error states of a system after knowing the initial measurement states and noise. Its theory of operation is based on the least mean-variance estimation theory. After assuming that linear models express the system, the measured noise and the noise corrupting the system are white noise, and the initial state vector is expressed as a random vector [17,18,21,35].

The KF process is divided into two states. The first state is prediction (time update) which is the current state of the system and its covariance estimates. The second state is a correction (measurement update) in which the KF obtains the measurements from the prediction state and measurements from an aided source (GNSS). Moreover, the KF calculates the Kalman gain $(K)$ which minimizes the mean square error of the estimates and updates its covariance estimates to utilize these updates to update the last estimates to eliminate the errors, as shown in Figure 7 [20,36].

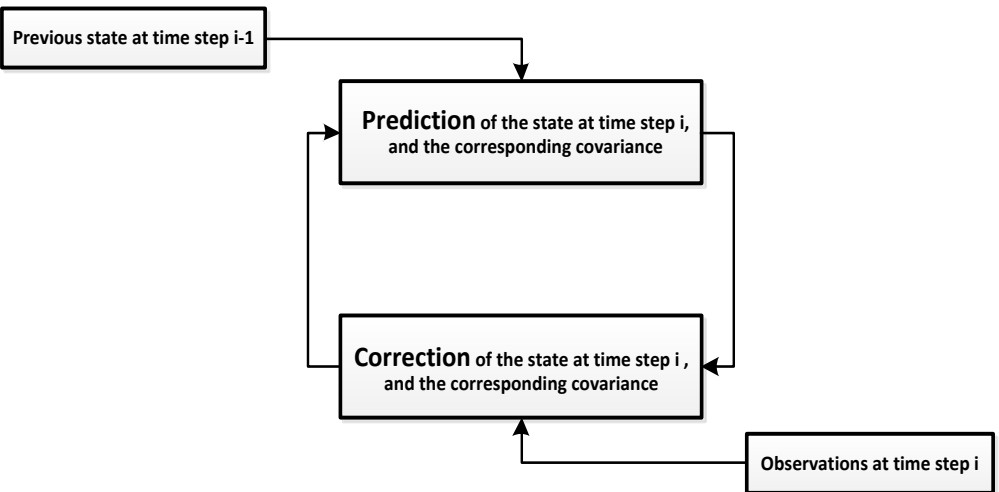

**Figure 7.** Kalman filter cycle.

The extended KF (EKF) is considered as a closed-loop configuration of the KF, where the KF obtains the error states and feeds back them to the INS algorithm to predict a more accurate INS solution and keep the system model in the linearity region [37–39].

The EKF deals with non-linear systems and utilizes Taylor expansion as a linearizing technique by taking only the first order of the expansion. As long as the higher-order terms of the expansion are very small and undervalued, the EKF obtains a more accurate estimation than the KF. The EKF has the disadvantages of its complexity and the requirements of the Jacobean calculations, which are very hard to obtain [35]. Both loosely coupled and tightly coupled integration are utilized in the paper. First, the KF used for this paper operates in a loosely coupled fashion to fuse the GNSS positions and velocities with the 2D RISS computed position and velocity components. A block diagram of the 2D RISS and GNSS integration is shown in Figure 8.

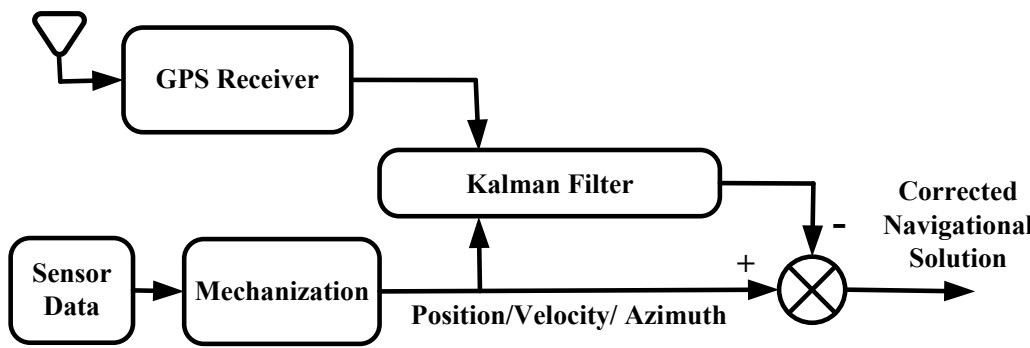

**Figure 8.** Schematic diagram of the Kalman filter for RISS/GNSS Integration.

The KF fuses the RISS computed position and velocity components with the corresponding GNSS positions and velocities when the RISS-based loosely coupled integration approach is used. This enables the computation of the positions, velocities, and altitude errors as well as sensor errors. When a GNSS outage occurs (i.e., less than four satellites are visible to the receiver with clear line-of-sight), the KF will only run the prediction stage of the filter and relies mainly on the error model.

## 8. PCI for Modeling Azimuth Errors

While using the low-cost MEMS-based inertial sensors, the application of KF linear error models with stationary white Gaussian noise for error state estimation can lead to quick deterioration of the navigation solution during GNSS outages due to their composite

error characteristics. For RISS, residual azimuth errors after KF prediction of the linear part of these errors were the principal cause for the deterioration of the solution. PCI, a system identification technique that can be utilized to model the residual azimuth errors, can overcome the limitation of Kalman for RISS/GNSS integration and can increase the performance.

When GNSS is available, KF is employed to perform RISS/GNSS integration. In parallel, as a background routine, the prediction of the KF azimuth is used together with mechanization results and the GNSS aiding azimuth to derive the true non-linear residual error of the azimuth. The block diagram that shows RISS/GNSS integration and includes the identification of the non-linear azimuth error by PCI is shown in Figure 9. The training data provided the reference output to construct the azimuth residual non-linear error PCI model. Moreover, the KF sent azimuth predictions to PCI as the input to build the model. The input and output system dynamics help to identify non-linear errors, and the algorithm can then achieve a residual non-linear azimuth error model.

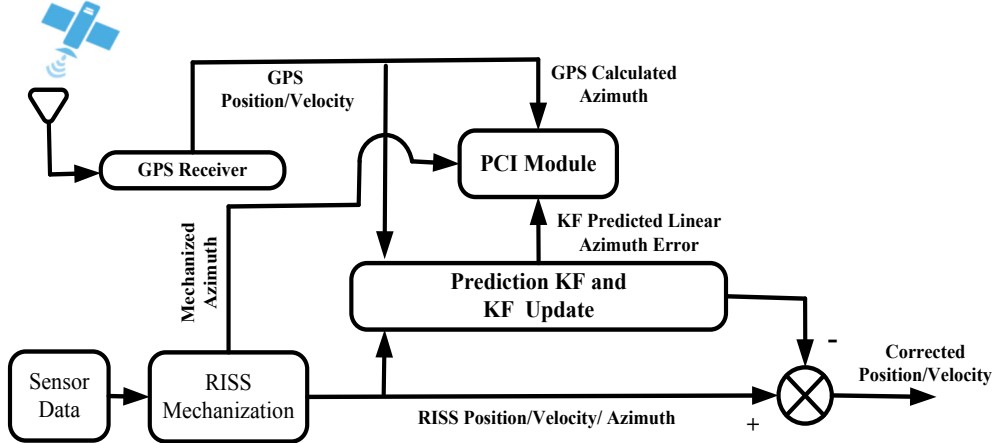

**Figure 9.** Loosely coupled KF-PCI technique during GNSS availability.

When less than four satellites are visible, GNSS outage occurs, as a loosely coupled architecture is used. When there is a GNSS outage, the identified parallel cascade will be utilized to predict the azimuth errors (residual and non-linear) from the KF prediction for the linear azimuth error. After correction, the azimuth angle is passed to the new mechanization shown in Figure 10 to calculate the corrected position and velocity.

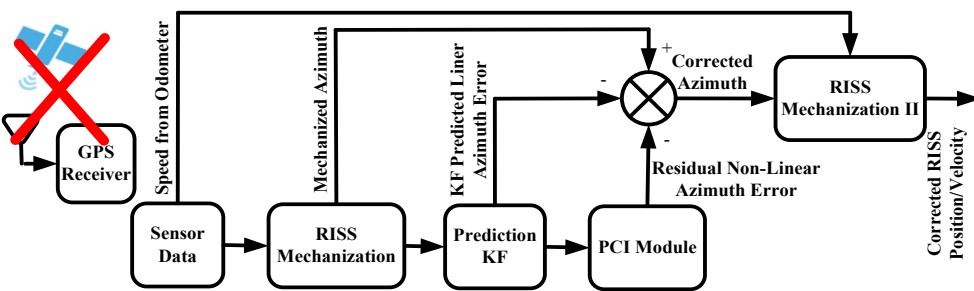

**Figure 10.** Loosely coupled KF-PCI technique during the GNSS outage.

A road test trajectory using ultra-low-cost ADI IMU and the low-cost Trimble Lassen SQ GNSS receiver was conducted in Kingston, ON, Canada, for nearly 2000 s. The NovAtel ProPak-G2-Plus combines a GNSS receiver, and the Honeywell HG1700 IMU via SPAN technology is used as a reference navigation solution. The specifications of the utilized IMUs can be found in Table 1.

**Table 1.** Performance characteristics of imus.

| IMUs | ADI (100 HZ) | HG1700 IMU (100 HZ) | IMU-CPT (100 HZ) |
|---|---|---|---|
| Size (cm$^3$) | $7.62 \times 9.53 \times 3.2$ | $19.3 \times 16.7 \times 100$ | $15.2 \times 16.8 \times 8.9$ |
| Weight | 0.59 Kg | 3.4 Kg | 2.28 Kg |
| Max data rate | 100 Hz | 100 Hz | 100 Hz |
| Start-up time | <1 s | <5 s | <5 s |
| **Accelerometer** | | | |
| Range | $\pm5$ g | $\pm50$ g | $\pm10$ g |
| Bias instability | $\pm6$ mg | $\pm1$ mg | $\pm0.75$ mg |
| Scale factor | <0.2%, $1\,\sigma$ | 300 ppm, $1\sigma$ | 300 ppm, $1\,\sigma$ |
| **Gyroscope** | | | |
| Range | $\pm150\,°/s$ | $\pm1000\,°/s$ | $\pm375\,°/s$ |
| Bias instability | $<\pm0.5\,°/s$ | $1.0\,°/h$ | $\pm1.0\,°/h$ |
| Scale factor | <0.1 %, $1\sigma$ | 150 ppm, $1\sigma$ | 1500 ppm, $1\,\sigma$ |

Eight simulated GNSS outages of 120 s each were introduced in post-processing for several vehicle dynamic conditions, including high speeds, slow speed, turns, straight portions, and stops, as shown in Figure 11.

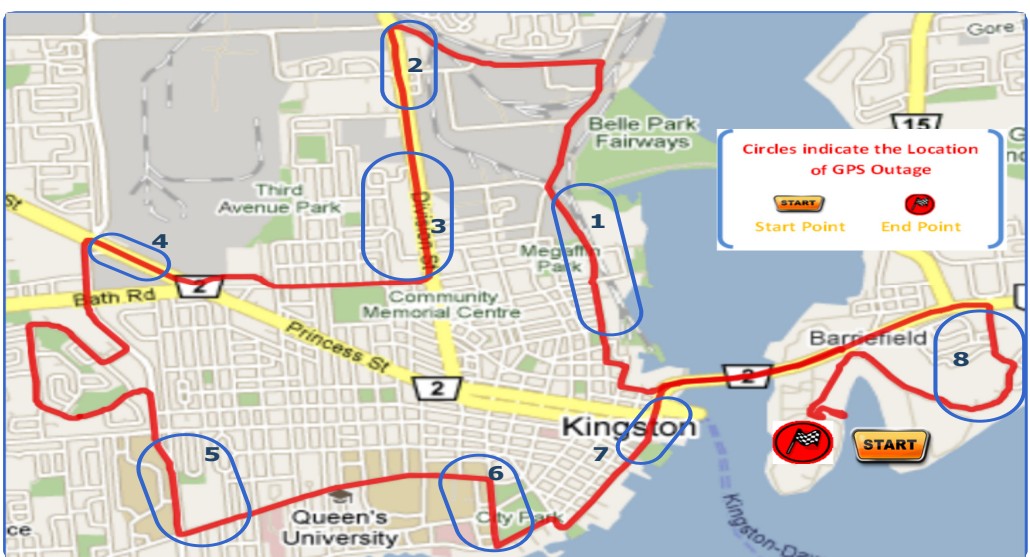

**Figure 11.** Road test trajectory and circles indicate the approximate locations of 120 s GNSS outages.

The errors of KF-PCI and KF-only solutions were compared with respect to the NovAtel reference solution.

The comparison of KF-PCI and KF-only solutions for RISS/GNSS integration is presented in Table 2. The system identification technique PCI, along with KF, was able to model and diminish the residual and non-linear errors in the azimuth and improved the results for eight simulated GNSS outages by 77.91%.

**Table 2.** Two-dimensional position rms-error comparison.

| Outage No. | Outage Dur. (s) | RMS Error in Position (Meter) | |
| --- | --- | --- | --- |
| | | KF | KF-PCI |
| 1 | 120 | 20.3 | 11.1 |
| 2 | 120 | 10.7 | 6.3 |
| 3 | 120 | 17.6 | 8.8 |
| 4 | 120 | 17.8 | 7.9 |
| 5 | 120 | 35.9 | 10.9 |
| 6 | 120 | 84.2 | 8.5 |
| 7 | 120 | 62.9 | 16.5 |
| 8 | 120 | 91.4 | 7.9 |
| Average | | 42.6 | 9.7 |

## 9. PCI for Enhancing KF Based Tightly-Coupled Navigation Solution

For loosely coupled integration, a clear line-of-sight between the receiver and no less than four satellites is considered a prerequisite to provide position, velocity, and timing aiding. The signals transmitted by the GNSS satellites can suffer from frequent interference and signal blockage in urban canyons and thick foliage where an uninterrupted clear view of the sky for the receiver is not presumable. Tightly coupled integration using the 3D reduced inertial sensor system is a better choice in challenging GNSS scenarios, especially when the number of visible satellites is three or less, as it can provide GNSS aiding. However, errors of pseudoranges measured by the GNSS receiver used as aiding in the RISS/GNSS integrated solution will affect the overall positioning accuracy. This section of the paper explores the benefits of using PCI, a system identification technique for modeling residual pseudorange correlated errors that can be utilized by a KF-based tightly-coupled RISS/GNSS navigational solution. PCI can improve the overall navigation solution by modeling residual pseudorange correlated errors to be used by a KF-based tightly-coupled RISS/GNSS navigational solution, as shown in Figure 12.

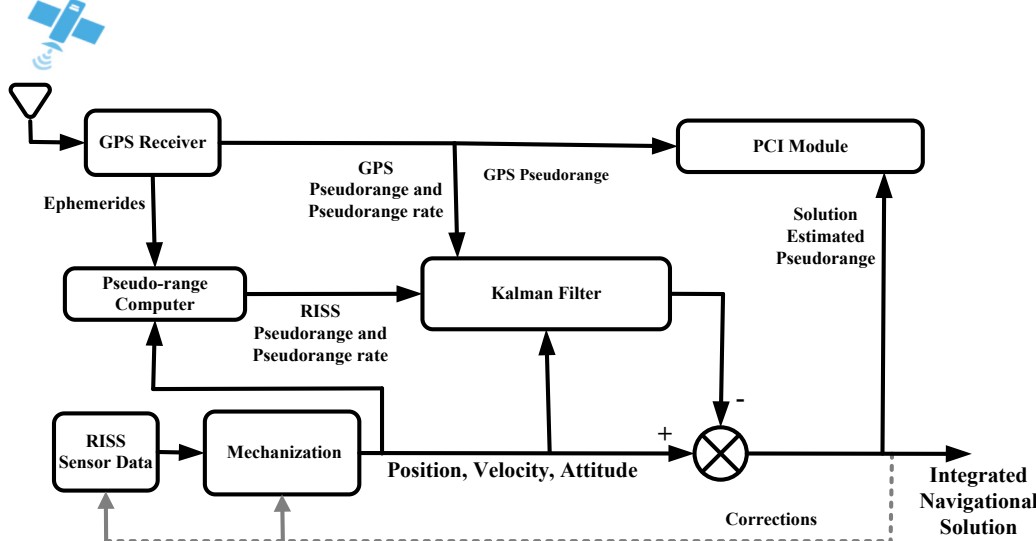

**Figure 12.** Block diagram of non-linear system identification to model the pseudoranges during GNSS availability using tightly coupled KF.

When less than four satellites are visible, the PCI model for the visible satellites is utilized to estimate the residual pseudorange errors for these satellites, and the corrected pseudorange value is provided to the KF, as shown in Figure 13.

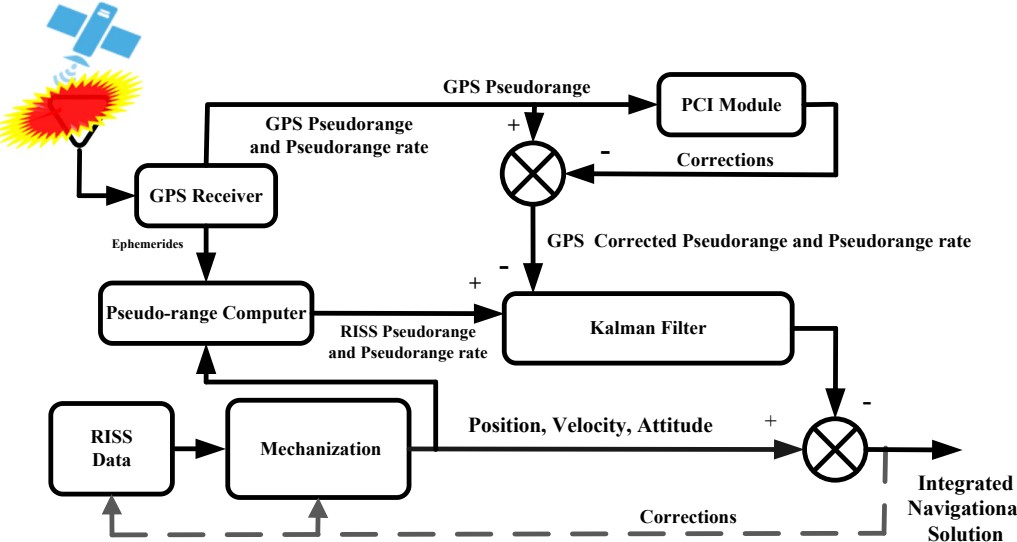

**Figure 13.** Tightly coupled KF-PCI technique during partial GNSS outage.

Around a 3000 s long trajectory was considered to check the validity of the proposed technique. It started at the Royal Military College of Canada, covering the major roads in the city of Kingston. Six 60 s GNSS outages were introduced in post-processing during good GNSS availability, as shown in Figure 14 on the map as blue circles.

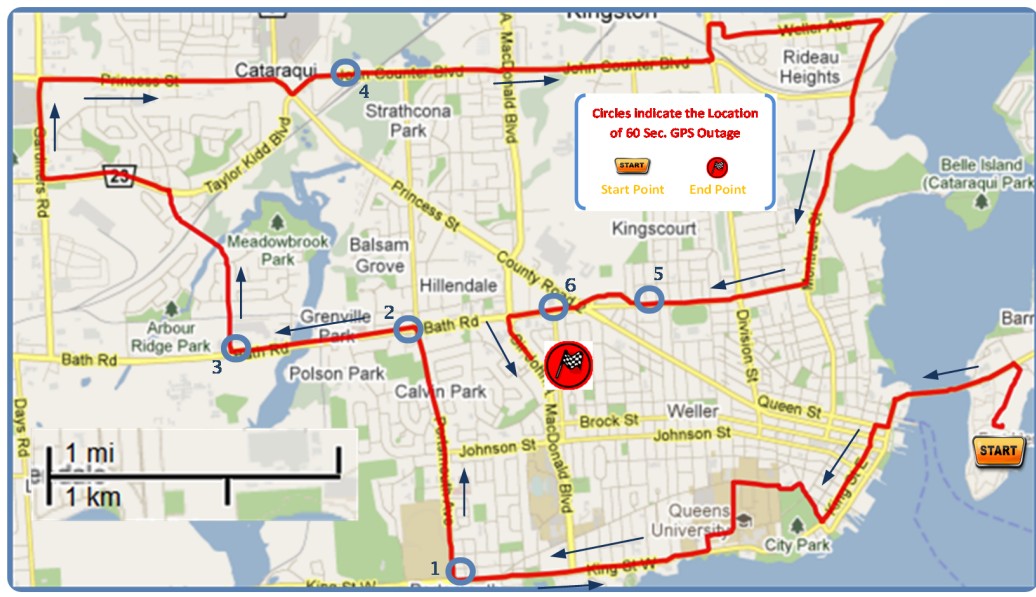

**Figure 14.** Road test trajectory and circles indicate the approximate locations of 60 s GNSS outages.

The trajectory was tested by partial outages having 3, 2, 1, and 0 visible satellites, respectively. The errors estimated by KF-PCI and KF-only solutions for RISS/GNSS integration were evaluated with respect to the NovAtel reference solution. Table 3 shows the average RMS position errors in meters.

**Table 3.** RMS horizontal position error over all partial GNSS outages.

| The Number of Visible Satellites | Outage Dur. (s) | RMS Error in Position (Meter) | |
|---|---|---|---|
| | | KF | KF-PCI |
| 3 | 60 | 7.5 | 4.6 |
| 2 | 60 | 10.4 | 8.7 |
| 1 | 60 | 15.8 | 15.7 |
| 0 | 60 | 15.6 | 15.6 |

The most significant performance of the PCI build model for pseudoranges error corrections was observed when three satellites were available since three corrected ranges served the tightly coupled solution, offering the highest effect. For RMS position errors, the performance enhancement of the KF-PCI over KF-only solution was 38.68%. The contributions of pseudorange error corrections using PCI continue to diminish for the cases of two satellites and one satellite. For RMS position errors, the improvement using the proposed PCI model reduced to 16.48% for KF-PCI over KF-only for two-satellite cases. There was no improvement using the proposed PCI model for one-satellite cases. No corrections were available for the PCI build model for pseudorange errors in the case of zero satellites, and the solutions provided by KF-PCI and the traditional KF were equivalent.

## 10. Conclusions

This paper has discussed PCI, a non-linear system identification technique to improve the performance of the integrated RISS/GNSS system. Two versions of RISS were used, one based on the single-axis gyroscope, along with an odometer, proposed by the author, integrated with GNSS, and the other incorporating two accelerometers to calculate pitch and roll. The complementary strengths of GNSS and RISS can be synergized, and optimal performance would be achieved during GNSS outages. First, loosely coupled and then tightly coupled integration schemes were considered. Enhancements for both integration techniques were suggested, successfully implemented, and tested for real road trajectory data using KF. As demonstrated by the results, PCI can handle the gyroscope's non-linear azimuth errors and stochastic sensor errors for the first algorithm. The performance of the 2D positioning RMSE KF-PCI solution was improved by about 77% as compared to the KF solution for the multiple GNSS outages of 2 min. For the second algorithm, PCI is utilized for modeling the residual pseudorange correlated error. In the case of the availability of three satellites, the performance improvement for the proposed KF-PCI solution was approximately 39% compared to the KF solution for the one-minute partial outage. The results demonstrated the worth and effectiveness of the proposed system identification techniques for enhancing the integrated navigation system at various phases.

**Author Contributions:** U.I., A.A., and J.G. proposed the idea, conceived and designed the algorithm, performed the experiments, and wrote the manuscript; A.U. participated in the experimental work, analyzed results, and reviewed the simulation results; A.N. and M.J.K. supervised the activities, reviewed the manuscript, and provided important suggestions to improve the algorithm. All authors have read and agreed to the published version of the manuscript.

**Funding:** This work was supported by the Natural Sciences and Engineering Research Council of Canada (NSERC).

**Institutional Review Board Statement:** Not Applicable.

**Informed Consent Statement:** Not Applicable.

**Data Availability Statement:** Not Applicable.

**Conflicts of Interest:** The authors declare no conflict of interest.

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
