# Peer review of "Implementation of Parallel Cascade Identification at Various Phases for Integrated Navigation System"

_futureinternet, doi:10.3390/fi13080191_

Round 1

Reviewer 1 Report

Dear Editor,

my comments for Authors:

  1. abstract, all acronyms must be explained in this chapter.
  2. abstract, please also add one sentences with obtained results from research test.
  3. main body of text, all acronyms must be explained also.
  4. chapter 2, please add one sentences about application GNSS in aerial navigation, because I don't see it. Please add the example of papers from MDPI journals.
  5. algorithm (1-17), please check if all symbols are explained in the text.
  6. Figure 10 is illegible, please correct it.
  7. Figure 11, please improve the quality of this Figure
  8. Figure 12, please improve the quality of this Figure
  9. Figure 13 and 14 are illegible, please correct it.
  10. Figure 15, please improve the quality of this Figure
  11. conclusion, please write about your findings from research test, please also add the obtained results. Why your methos is better than another solution?
  12. References, see comment 4.

Author Response

We want to thank the reviewers and editor for their objective and thorough review of our paper and valuable comments. We have addressed the reviewers' and editor's comments in the following point-by-point response, and the manuscript has also been changed accordingly. All the changes made to the original manuscript to address the reviewers' and editor's comments are highlighted in red in the revised manuscript.

Reviewer 2 Report

-Submitted in "Future Internet"?

Abstract:
- try not to put acronyms on the abstract (MEMS), (INS) etc.
- Maybe "GPS" could be replaced by "GNSS" as denied environment concerns all constellations.

-l51: uppercase for PCI acronym

-Introduction: First mention of "error models" at the very end of the introduction. Maybe introduce why error models are important to estimate for the overall navigation accuracy.

-l56 The position of the antenna is computed by a GNSS system, not the receiver (unless lever-arm compensated).

l59-62: Not clear

-l70: Spoofing as well

-l75: GPS or GNSS environment?

-l83: 85? and upper case "The"

-l87: "Integrated in a strapdown manner" instead of "transformed" maybe?

-l101: some error models can be added in an augmented state vector to estimate, bias, AR1 process, random walk, etc.

-l205: "RISS" already explained

-l208-9: is b_z  Gyro bias?

-eq(12): I don't understand the sign -(). Do you rotate from or to body-local reference?
I might be wrong: if the ODE (Adot) for azimuth is expressed in the local frame, the gyro reading from the inertial to the body frame, expressed in the local frame omega_ib^l (+ bias) needs first to be rotated in the local frame (omega_z is a rotation from inertial to the body, expressed in the body frame). Then it can be subtracted with the two other components (rotation and curvature) which are expressed already in the local-reference frame?

-l201: curiosity: Are you sure that using instantaneous accelerometer readings minus an odometer to compute pitch and roll angle, with the determination of g (either model or norm of static 3 orthogonal accelerometers which you don't have apparently (missing f_z)) with a driving platform (bumps) adds less noise than integrating gyro readings to obtain attitude? Why bother about the Earth curvature and rotation which are so small if raw MEMs accelerometer measurements are used to compute attitude in this manner?

7. Kalman Filter

(E)KF is widely used and known, I wouldn't explain it too much as the paper is already long.
-l248: Figure 7 and Figure 8 are nice, but redundant in my opinion.

Figure 9: This scheme is a bit confusing. The "Sensor Data" mechanization (Strapdown) gives you the predicted X_k- which will be corrected/updated with the GNSS measurements in the update phases giving X_k+. The "addition" of both is done within the filter, not outside.

8. PCI For modeling Azimuth Errors

-l301: What is the accuracy of the NovAtel reference?
-Figure 12: Is it a 60 or 120 seconds GNSS outage?
-l304: "Table 1"

-Table1: Results impressive :-). Have you tried to model the gyro_z bias as the augmented state with the EKF and compare it without?

9. PCI for enhancing KF...

-l334: "Table 2".
-so KF performs better than KF-PCI (RMSE KF< KF-PCI in Table 2)? Explanation?

-l339: "os"

Author Response

(The authors gave the same response as above.)

Round 2

Reviewer 1 Report

I accept the paper.